# The Prevalence and Awareness Concerning Dietary Supplement Use among Saudi Adolescents

**DOI:** 10.3390/ijerph17103515

**Published:** 2020-05-18

**Authors:** Hanan Alfawaz, Nasiruddin Khan, Alwateen Almarshad, Kaiser Wani, Muneerah A. Aljumah, Malak Nawaz Khan Khattak, Nasser M. Al-Daghri

**Affiliations:** 1College of Food Science & Agriculture, Department of Food Science & Nutrition, King Saud University, Riyadh 11495, Saudi Arabia; a.f.m@windowslive.com; 2Chair for Biomarkers of Chronic Diseases, Biochemistry Department, King Saud University, Riyadh 11451, Saudi Arabia; wani.kaiser@gmail.com (K.W.); malaknawaz@yahoo.com (M.N.K.K.); aldaghri2011@gmail.com (N.M.A.-D.); 3College of Applied and Health Sciences, Department of Food Science and Human Nutrition, A’ Sharqiyah University, Ibra 400, Oman; knasiruddin@asu.edu.om; 4College of Medicine Medical Student, Almaarefa University, Riyadh 11597, Saudi Arabia; Aljumahmd@gmail.com

**Keywords:** dietary supplements, school student, adolescents, physical activity, awareness, Saudi Arabia

## Abstract

Current dietary supplement (DS) use among Saudi school students is not well described. In this study, we aim to investigate the prevalence and predictors of DS use among adolescents. This cross-sectional study collected data via self-administered questionnaire from 1221 students (12 to 18 years). The overall prevalence of DS use was 26.2%, significantly higher in females than males (33% vs. 17.9%, *p* < 0.001). High proportion of female DS users demonstrated normal BMI than males (84% vs. 56.5%, *p* < 0.001). High percentage of male DS users were engaged in vigorous and/or high physical activity (PA) levels than female DS users (58.2% vs. 43%, *p* = 0.022; and 57.1% vs. 20.7%, *p* < 0.001, respectively). The main reasons for DS use among females were vitamin deficiency (63.3%), hair condition (37.6%) and nail health (23.5%) while in males, the main reasons were vitamin deficiency (58.4%) and body building (34.4%). The predictors of DS use in Saudi adolescents included being female, having high family income and being physically active. In conclusion, the overall prevalence and preference of DS use, though low among Saudi adolescents, was driven mostly by gender, physical activity levels and socioeconomic factors like family income. DS use guidelines and counselling among Saudi adolescents are warranted to improve public health.

## 1. Introduction

An adequate diet plays a key role in the maintenance of good health and ensures sufficient amount of nutrients, vitamins and minerals in human body [1]. The consumption of dietary supplement (DS) has increased tremendously in the last few decades in all age groups including young generation with its use in promoting different health benefits, adequate diet intake and improving physical performance [2,3]. Based on European Food Safety Authority (EFSA), the DS are defined as “concentrated sources of nutrients or other substances with a nutritional or physiological effect intended to supplement a normal diet” [4]. European Union countries were the first to adopt dietary supplement regulatory act (Directive 2002/46/EC) [5]. In addition, the Dietary Guidelines for Americans (DGA) states that in certain cases, the use of fortified foods and DS may be favorable for health [6]. Studies demonstrate that DS users are more likely to adapt healthy lifestyle including balanced dietary patterns, regular exercise, achieve healthy body weight and avert harmful products such as tobacco, than non-users [7,8]. About half of the United States (US) adult population uses some form of DS including vitamins, minerals and plant products [9]. DS are available in various forms such as pills, capsules, tablets and powder containing vitamins, minerals, herbs, amino acids and other substances [10]. However, a considerable proportion of general population consume DS without having enough knowledge and avoid consulting or seeking advice with health professionals [11,12].

Based on data collected from NHANES, around 33.2% of children and adolescents in the US are using DS that mainly includes multivitamins, nutritional products (e.g., iron, calcium and vitamin D) and alternative medicines (e.g., body building supplements, melatonin and ω-3 fatty acid supplements) [13]. DS use ranges from 16% in Slovenia (children) to 45% in Finland (in adolescents) [14,15]. While other countries such as Germany, Serbia and Malaysia have demonstrated considerable use of DS in young population (21%, 68% and 43%, respectively) [16,17,18]. A Japanese survey reported that 20.4% of children and adolescents (<18 years) were using supplements or had used them in the past year [19]. The prevalence of DS use in Australian adolescent and children were 20.1% and 23.5%, respectively [20]. The existence of varied prevalence rate among different population indicates the inevitable effect of cultural and environmental factors in the use of DS [21]. The use of iron and folic acid supplementation by women in developing countries shows that cultural factor and belief plays a potential role in DS use [22].

The consumption of DS is affected by socio-demographic and lifestyle factors such as age, gender, healthy diet habits, level of education, income and physical activity. However, these factors was studied mostly in adult population and their association with DS consumption in children and adolescent is limited and inconsistent [7,8,23,24]. However, since age is one of the predictor of DS use, the prevalence rate in adults does not correspond to children and adolescents [7,22]. Young people are particularly at high risk of poor nutrition leading to nutrient deficiencies [25]. The factors which determine DS use in children and adolescent may be entirely different. The DS use in children is primarily based on their parents’ decision who believe that their children will not get balanced and sufficient nutrients from their regular diet, while adolescents usually decide by themselves based on information from social media or follow the opinion of their peer or coach [14].

The Kingdom of Saudi Arabia is among one of the fastest growing economy in Gulf Cooperation Council countries (GCC) that has affected its population via nutritional transition and changes in daily lifestyle behaviors leading to an increasing demand for nutritional supplements. There are studies about use of DS and multivitamins in different parts of Saudi Arabia among population of different age. A recent study reported a high prevalence of DS use among young females (18 to 25 years of age) than males (85.9% vs. 13.9%) [26]. Studies among university and college female students (22, ≥19 years of age, respectively) also demonstrated a high prevalence (67.7% (occasionally), 32.3% (regularly) and 76.6%, respectively) of DS use [27,28]. However, most of these studies were either gender specific, or included male and female university students (≥18 years of age). Our primary objective was to explore the gender-based prevalence of DS use among middle and high school students aged 12 to 18 years; and demonstrate the relationship between anthropometric, demographic, socioeconomic and physical activity with DS use. In addition, the study aims to observe the reason, source and awareness of DS use among the selected population.

## 2. Materials and Methods

### 2.1. Study Design

This study was conducted in collaboration with Ministry of Education and according to Noor program—an online program that connects all schools to the ministry and provides full functionality of school administration for all schools in Saudi Arabia—population size was 373,842 students (male and female) in both private & governmental schools.

This descriptive, cross-sectional study was conducted from April to May 2018. Participants were recruited from both private and public middle and high schools, chosen randomly from all regions (north, south, east, west and center) of Riyadh. The exclusion criteria were occurrence of a disease requiring a special diet, pregnancy or lactation and incomplete questionnaire. Consent form was obtained from Dean of Scientific Research King Saud University, Riyadh Saudi Arabia and all subjects gave their written informed consents for inclusion before they participated in the study. This study was conducted according to the guidelines laid down in the Declaration of Helsinki. The study design and protocol was approved by the Ethics Committee for Scientific Research and Post Graduate Studies at the College of Science, King Saud University, Saudi Arabia (reference# KSU-HE-18-257).

### 2.2. Study Population, Data Collection and Measurements

In this cross-sectional study, 1221 participants (552 males, 669 female) of 12 to 18 years of age completed a self-administered questionnaire about DS. The questionnaire included various sections such as sociodemographic, anthropometric, PA, reason, source, form and general awareness about DS use. A pilot study (100 student) was performed to confirm the reliability and validity of the questionnaire. Cronbach’s α, an estimate of coefficient was measured for the questionnaire and the value obtained was excellent (84%). Experts in the related field reviewed the questionnaire as well and adjustments were made to strengthen the reliability. We distributed the questionnaire among students to get their feedback ensuring they have understood all questions. A professional dietitian accompanied the participants during filling of the questionnaire to clarify any doubt or misunderstanding. After the completion, the questionnaire was revised and collected for further use.

Sample size calculation was based on Raosoft online calculator, which is a calculation used to specify the sample size and locate how many responses are needed with an error margin to meet the desired confidence level. The total population of students in Riyadh is approximately 373,842 and in order to obtain a confidence level of 95% and a 2.8% margin of error, a minimum sample size of 1221 would enable us to achieve the study objectives.

The questionnaire contained five (5) different sections:DS useAnthropometry and sociodemographic factors including age, marital status and education level. The family income was divided depending on the amount of Saudi Arabian Riyal (1 SAR = 0.266 USD): higher income class (>16,000), upper income class (10,001–16,000), middle income class (5000–10,000); and low income class (< 5000). The body mass index (BMI) was calculated from body weight and body height and divided into three categories (normal, overweight and obese) based on age and sex specific cut-offs for children [29].The extent of PA among adolescent was based on the level of physical activity over the past week, determined with an adapted school health action, planning and evaluation system PA electronic questionnaire [30]. Average daily PA (in hours) was then calculated according to Wong and Leatherdale [31], as was average daily energy expenditure for PA (kcal/kg/day), which was used to classify participants as light PA, moderately active, high and vigorously active. The categories of sports and activities included in different PA level were: Vigorous physical activity (running, bicycle riding, weight lifting, swimming laps), High physical activity (tennis, handball, soccer, basketball), Moderate physical activity 1 (brisk walking, house cleaning, gymnastics), Moderate physical activity 2 (table tennis, house arranging, dancing) and light physical activity (walking, ironing, washing dishes). The frequency of exercise as (daily, three or four times per week, one to two times per week, few times a month and once per month), gym membership and its duration was also included in the questionnaire.The frequency of awareness about DS use.Reason, form and source for DS use.

### 2.3. Statistical Analysis

Data were analyzed using the Statistical Package for Social Sciences (SPSS) 22.0 (SPSS, Inc., Chicago, IL, USA). All categorical variables were presented as frequency and percentages (%) and the difference among genders was checked using chi square and Fisher exact test. Continuous data were presented as mean ± standard deviation (SD). A Cronbach’s alpha test was used to check questionnaire reliability (Cronbach’s alpha > 0.70). All continuous variables were checked for normality using Kolmogorov–Smirnov test. An independent *t*-test was used to check mean difference for continuous variables age, etc.

To examine the strong predictors of DS use, the binary (crude data) as well as multivariate-adjusted odds ratios (ORs) with 95% confidence intervals (95% CIs) were calculated using logistic regression models. All models were mutually adjusted for all potential confounders. The multivariate-adjusted models included potential determinants of DS use: gender, BMI, education level, family income level, physical activity including various categories and marital status. *p* value <0.05 was considered statistically significant.

## 3. Results

### 3.1. Prevalence of DS Use

DS use among out study participants is presented in Table 1. The overall prevalence of DS consumption among our participants was 26.2%. DS use was significantly high in females than males (33% vs. 17.9%, *p* < 0.001).

### 3.2. Sociodemographic Characteristics and Anthropometrics

Table 2 represents anthropometric and socio-demographic characteristics of our study population. The total number of participants (N) were 1221 with an overall age of 15.8 ± 1.6 years including 552 males and 669 females. Based on BMI, most of the DS users fall in normal range (75.3%) followed by overweight (17.5%) and obese (7.2%). However, the percentage of females in normal BMI category was significantly high than males (84% vs. 56.5%, *p* < 0.001), respectively. In addition, most of the DS users were unmarried (94.6%). Irrespective of gender, a statistically insignificant increase in proportion of DS users was observed with increase in family income level. The prevalence of DS use was not associated with education level among participants.

### 3.3. Level of Physical Activity (PA) among Dietary Supplement Users

Table 3 presents the physical activity (PA) levels among male and female dietary supplement users. Significantly high percentage of males compared to females were involved in vigorous and high PA level (58.2% vs. 43%, *p* = 0.022; 57.1% vs. 20.7%, *p* < 0.001), in contrast, the proportion of females were significantly high than males in moderate (1 and 2) levels of PA (46.6% vs. 34.1%, *p* = 0.054; 47.2% vs. 20.9%, *p* < 0.001), respectively. A significantly high percentage of males had previous gym membership rather than females (28.6% vs. 20.2%, *p* < 0.001). Although statistically insignificant, the present gym membership of the males was higher (31.6%) than females (9.4%). The duration of gym membership was higher in males than females for all duration range (1 to 3 months (31.3% vs. 13.6%), 3 to 6 months (14.1% vs. 4.1%), >6 months (16.2% vs. 7.2%), respectively). However, the males were significantly higher in discontinuing the gym membership for less than a month than females (7.1% vs. 6.3%, *p* < 0.001).

### 3.4. Knowledge and Awareness about DS Use among Study Subjects

Table 4 summarizes the awareness about DS use in our participants. A significantly high percentage of females rather than males responded correctly for majority of the questions about DS use such as, “Too much vitamin intake is harmful”? (Correct answer “yes”, 54.8% vs. 39.4%, *p* = 0.04); “Dietary supplement may substitute the food intake”? (Correct answer “No”, 84.6% vs. 71.7%, *p* = 0.009); and “Is periodic blood test necessary”? (Correct answer “yes”, 86.4% vs. 69.7%, *p* = 0.04), respectively. The percentage of females were also high for other such questions (as shown in Table 4) than males, but the differences were statistically insignificant. On the other hand, significantly high percentage of males than females responded correctly for the question, “DS are essential for healthy hair”? (Correct answer “No”, 26.3% vs. 5.4%, *p* < 0.001), respectively. 

### 3.5. Reason, Source and Form of DS Use

Table 5 represents the reasons, form and source for purchasing DS among participants. Among all reasons, significantly high percentage of females than males used DS due to doctor’s prescription, vitamin deficiency, hair condition and nail health (63.3% vs. 47.5%, *p* < 0.006; 37.6% vs. 17.2%, *p* < 0.001; 23.5% vs. 13.1%, *p* < 0.021), respectively, while males used more DS than females (34.4% vs. 9.5%, *p* < 0.001) only for body building purpose. The overall preferred form of DS intake among participants were pills (45%), followed by capsule (17.8%), liquid (11.9%) and powder (6.3%). However, compared with females, significantly high proportion of males (2.3% vs. 15.2%, *p* < 0.001) chose powder as their preferred choice for DS intake. The sources for DS purchase among participant were pharmacy (39.1%) followed by Internet (10.6%) and social media (2.2%). A significantly high percentage of females preferred to buy DS from pharmacy (42.5%, *p* = 0.034) while male chose to purchase mostly via Internet (22.2%, *p* < 0.001).

### 3.6. Determinants of DS Use

Table 6 represents the results of multivariate-adjusted odds ratio (ORs) demonstrating the independent predictors of DS use among adolescent. The significant strong predictors were being female, family income level and high level of physical activity. The DS users were less likely to be male (OR: 0.45, 95% CI: 0.32–0.63). The odds of DS use were 2.44-fold (95% CI: 1.11–5.33) higher in adolescents with high family income level. Adolescents engaged in PA had slightly increased odds of DS use, but the 95% CI included 1. Moreover, adolescents who do not perform high level of physical activity were less likely to be DS users (OR: 0.62, 95% CI: 0.44–0.86). The overweight adolescents were less likely to be DS users than normal weight, but again its CI included 1. Compared to adolescent with primary education, those with secondary educational had a slightly higher probability of DS use, but it is not significant.

## 4. Discussion

The first of its kind, our present study demonstrated a relatively low prevalence of DS use (26.2%) among school students. However, the prevalence was high among females rather than males. Moreover, the study indicated strong predictors of DS use as being female, high family income level and high level of physical activity.

As compared to US (33.2%) [13], the prevalence of DS use in our participants were low (26.2%). Moreover, the prevalence of DS use was far low as reported in adolescent population from different countries such as Malaysia (43%) [18], Finland (45%) [15] and Serbia (68%) [17]. On the contrary, the prevalence was a bit high than German [16] and Japanese adolescents [19] (21%, 20.4%, respectively). A recent study in Australian population demonstrated high use of DS in adolescent females (20.6%) than males (19.7%) [20]. In our study, the high prevalence of DS uses in adolescent females (33%) than males (17.9%) was in line with NHANES results demonstrating sex difference with more adolescent females (33.4%) using any DS than males (23.9%) [13].

Healthy lifestyle is a combined effect of balanced diet and physical activity. DS, itself is not a complete diet and compliments a normal diet for various reasons. DS is used to attain recommended intakes or minimize nutritional deficiencies. DS users make a greater effort to seek health and wellness than non-users. Our recent studies on adults reported significant positive effects on health outcomes like diabetes and metabolic syndrome in interventions that combined approaches like balanced diet and increased physical activities [32,33,34]. Such studies on children and adolescents in this region are limited. However, a study on health behavior in school-aged children (HBSC) database of 167,021 children and adolescents from 37 countries and regions [35] reports that a composite higher score of healthy lifestyle like daily physical activity, healthy diet (which included dietary supplements as it is hard to get complete nutritional value from diet alone), etc. presented a less likelihood of having multiple health complaints. This indicates that health outcomes are a combined effect of healthy lifestyle including balanced diet with DS use and physical activity.

The use of DS is associated with normal BMI in children as well as adolescent [36]. High percentage of females with normal BMI (67.1%) using DS than males (55.4%) were reported among general population in Riyadh, Saudi Arabia [37]. Among DS users, our results are consistent with these findings showing a significant high proportion of females (84%) with normal BMI than males (56.5%). Marital status was known as one of the factors influencing the DS use mostly among adults. Till now, the relationship of social status with DS use are inconsistent showing its use in unmarried [37], married [38], as well as no association [39]. College and university going single adolescents are more self-dependent, actively indulged in taking care of themselves and are more concerned about their health and physical image. Our result supports the above study [37] showing high percentage of unmarried DS users (94.6%) than married.

The quality of food and nutrient intake is likely to be influenced by income, with an observed low nutrient-dense food intake in people with low income level [40]. Moreover, a recent study in US demonstrated high DS use with increased income [41]. On gender basis, our finding showed no significant difference between DS use and family income. However, overall DS use among participants increased with an increase in family income, which was consistent with other similar studies performed in children as well as adolescent [42,43].

DS use in young males is mainly attributed to their interest in muscle building while females generally use DS due to aesthetic and health reasons [27,28,44]. In our study, majority of DS users were engaged in PA. However, females showed a significantly high percentage in moderate PA level than males, while males dominated in vigorous PA level than females. Similarly, study performed by Al-Hazzaa demonstrated high proportion of Saudi females engaged more in moderate PA than males, whereas males participated more in vigorous PA compared with females. The contributing factors were moderate-intensity activities such as carrying loads (e.g., babies), scrubbing floors, sweeping and vacuuming and use of stationary exercise machines at home [45].

The willingness of today’s young generation to improve body shape, gaining muscle and improve sport performance in short time is prompting towards increased use of DS especially among those who exercise in gym. Studies suggests a high prevalence of DS use in male gym users than female [46,47]. However, the information about DS use in gym users are primarily based on unreliable sources (Internet, fitness coach and friends) and usually self-prescribed [48]. Our findings showed a significantly high percentage of male DS users having gym membership (28.6% vs. 20.2%) than females. In addition, the duration of exercise in gym was high in males including all categories (less than month, 13 months, 36 months, >6 months) than females. Although insignificant, the frequency of exercise was highest among males on daily basis than females (33% vs. 25.5%, respectively). This finding is consistent with similar studies demonstrating high prevalence of DS use among male gymnasium user with more frequent and longer period of exercise rather than females [46,47].

Generally, females are more health conscious and the increased use of DS (such as calcium, multi-vitamin and minerals, vitamin D and omega 3) in their daily life may be attributed to maintain hair condition, maintain bone health, reduce bone injury, reduce illness and as a precautionary step to avoid osteoporosis later in their life [7,28]. Our study demonstrated more awareness in females about DS use than males. High number of females responded correctly than males for many questions such as “excessive use of vitamin intake is harmful to health”, “taking DS with your doctor’s instructions promotes body’s health” and “reading instructions for using DS is necessary” (in Table 4). A significant high number of females (84.6%) than males (71.7%) knew that DS cannot replace regular diet intake and blood test at regular interval helps to keep a check on our health status. As far as males were concerned, a significant high number of males (26.3%) responded correct for only one question (DS use is not essential for healthy hairs) than females (5.4%), showing scarcity and limited awareness among them. Similar findings were reported in other studies demonstrating more accurate knowledge about DS use among adolescent female students [20,27].

In today’s time, the information about DS use is easily available from many sources such as magazines, Internet, social media and television advertisements. Moreover, online purchasing is an increasing trend among young generation due to variety of options and affordable price. Females are more dependent on their family members for information about DS use and they prefer to buy from pharmacy [44]. Consistent with other studies, our study demonstrated a high proportion of females who prefer to buy DS from pharmacy (42.5%) than males (31.3%) [27,37]. In addition, a significantly high number of male students prefer to purchase DS via Internet (22.2%) which is consistent with similar observation conducted in Japanese college students [44].

In our study, gender differences for reasons of DS use were observed among students. As stated earlier, the main reasons for DS use in females were vitamin deficiency (63.3%) and aesthetic (to maintain hair condition (37.6%) and nail health 23.5%), while a significantly high number of males used DS for muscle building purpose (34.4%) [28,44,45,46,47]. The other reason to use DS in males were disease prevention (22.2%), but it was statistically insignificant than females (17.6%). These results were similar to those of other studies that reported similar gender-based reasons for DS use [28,37,44,45,49]. The preferred form of intake among majority of the DS users was pill (45%), but for unknown reason, a significantly high proportion of males preferred powder form for DS intake than females (15.2% vs. 2.3%). Consistent with some other studies [50], the findings in this study favor gender-specific intervention programs for adolescents to instill knowledge on correct and wise usage of DS. Moreover, a risk of misuse of DS among children and adolescents [51] warrant educational intervention for correct and wise usage and better public health.

There are few limitations that need to be considered while interpreting the results of the present study. First, since our study was based on random selection of schools, the sample size does not represent the overall adolescent population of all schools in Saudi Arabia. Second, our results demonstrate limited number of DS use predictors with possibility of missing some important factors of DS use to be taken into account. Overall, our findings suggest being female, high family income and high physical activity as predictors of DS use. However, the cross-sectional nature of this study was not able to specify the exact cause of DS use in adolescents.

## 5. Conclusions

A relatively low prevalence (26.2%) of DS use was reported among Saudi adolescents. Gender was the main predictor with majority of DS users being females having normal BMI than males. High level of physical activity was predictor of DS use among adolescents. The majority of the DS users were engaged in physical activity, with females mostly involved in moderate activity and males with vigorous activity level. Females exhibited high awareness about DS use rather than males. The gender-based reasons for DS use among females were physician’s prescription, vitamin deficiency and aesthetic; while males used it mainly for muscle building.

The present findings can be useful for developing health communications on DS use targeted to Saudi adolescents and monitoring future trends of supplement use. The indication of predictors about DS use in our study warrants further research in same section of population to identify any risk behavior and implement proper health education policies.

## Figures and Tables

**Table 1 ijerph-17-03515-t001:** Prevalence of dietary supplement use among participants.

Parameters	Overall (1221)	Males (552)	Females (669)	*p*-Value
Dietary Supplements				
Yes	320 (26.2)	99 (17.9)	221 (33.0)	**<0.001**
No	901 (73.8)	453 (82.1)	448 (67.0)	

Note: data represented as N (%). *p*-value significant at <0.05 level (bold).

**Table 2 ijerph-17-03515-t002:** Anthropometric, sociodemographic and lifestyle characteristics among dietary supplement users.

Parameter	Overall (1221)	DS Users (320)	Male DS Users (99)	Female DS Users (221)	*p*-Value
Age (year)	15.8 ± 1.6	15.8 ± 1.5	15.6 ± 1.8	15.9 ± 1.5	0.154
**BMI (kg/m^2^)**					
Normal	879 (72.6)	242 (75.3)	56 (56.5)	186 (84.0)	**<0.001**
Overweight	251 (20.6)	55 (17.5)	29 (29.3)	26 (12.0)	
Obese	91 (7.5)	23 (7.2)	14 (14.1)	9 (4.0)	
**Marital status**					
Married	34 (2.7)	11 (3.5)	9 (9.0)	2 (0.9)	0.068
Unmarried	1173 (96.1)	306 (94.6)	90 (91.0)	216 (98.1)	
Divorcee	14 (1.4)	3 (1.9)	0 (0.0)	3 (1.0)	
**Family income (monthly)**					
<5000 SAR	83 (6.8)	12 (3.7)	8 (6.8)	4 (2.2)	0.130
5000 to 10,000 SAR	172 (14.1)	40 (12.5)	14 (13.6)	26 (11.9)	
10,001 to 16,000 SAR	230 (18.8)	56 (17.5)	19 (19.3)	37 (16.8)	
>16,000 SAR	684 (56.1)	187 (58.8)	57 (59.1)	130 (58.9)	
No Data	52 (4.2)	25 (7.5)	1 (1.1)	24 (10.3)	
**Education level**					
Primary	486 (39.8)	123 (38.5)	43 (43.9)	79 (36.1)	0.116
Intermediate	735 (60.2)	198 (61.5)	56 (56.1)	142 (63.9)	

Note: data represented in N (%) for categorical variables and mean ± standard deviation for age. *p*-Value calculated for DS users in the two sexes. *p* significant at <0.05 level (bold).

**Table 3 ijerph-17-03515-t003:** Relationship between physical activity and dietary supplement use among participants.

Parameters	All (1221)	All DS Users (320)	Male DS Users (99)	Female DS Users (221)	*p*-Value
**Physical Activity (PA)**					
Yes	1023 (83.8)	284 (91.6)	91 (91.9)	193 (87.3)	0.156
No	198 (16.2)	36 (11.4)	8 (8.1)	28 (12.7)	
**If yes, types of PA?**					
Vigorous PA	448 (43.8)	136 (47.9)	53 (58.2)	83 (43.0)	**0.022**
High PA	408 (39.9)	92 (32.4)	52 (57.1)	40 (20.7)	**<0.001**
Moderate 1 PA	435 (42.5)	121 (42.6)	31 (34.1)	90 (46.6)	**0.054**
Moderate 2 PA	379 (37.0)	110 (38.7)	19 (20.9)	91 (47.2)	**<0.001**
Light PA	387 (37.8)	100 (35.2)	27 (29.7)	73 (37.8)	0.113
**How often do you exercise?**					
Once per month	103 (8.4)	17 (5.5)	3 (3.1)	15 (6.6)	0.163
A few times per month	192 (15.7)	51 (15.9)	11 (11.3)	40 (17.9)	
1–2 times per week	285 (23.4)	68 (21.4)	26 (25.8)	43 (19.3)	
3–4 times per week	287 (23.5)	90 (28.2)	26 (26.8)	64 (28.8)	
Daily	337 (27.6)	89 (27.8)	33 (33.0)	56 (25.5)	
Not Done	17 (1.4)	4 (1.3	0 (0.0)	4 (1.9)	
**Gym member?**					
Previous	266 (21.2)	73 (22.8)	28 (28.6)	45 (20.2)	**<0.001**
Present	160 (13.1)	54 (16.4)	31 (31.6)	21 (9.4)	
Never	795 (65.1)	195 (60.8)	40 (39.8)	156 (70.4)	
**Duration of Gym membership**					
Less than a month	69 (5.7)	21 (6.6)	7 (7.1)	14 (6.3)	**<0.001**
1 to 3 months	191 (15.6)	61 (19.1)	31 (31.3)	30 (13.6)	
3 to 6 months	75 (6.1)	23 (7.2)	14 (14.1)	9 (4.1)	
>6 months	115 (9.4)	32 (10.0)	16 (16.2)	16 (7.2)	
No membership	771 (63.1)	183 (57.2)	31 (31.3)	52 (68.8)	

Note: data represented in N (%) for categorical variables and mean ± standard deviation for age. *p*-value calculated for DS users in the two sexes. *p* significant at <0.05 level (bold).

**Table 4 ijerph-17-03515-t004:** Awareness about dietary supplement use among participants.

	All DS Users (320)			Male DS Users (99)			Female DS Users (221)			*p*
	Yes	No	Do Not Know	Yes	No	Do Not Know	Yes	No	Do Not Know	
DS are essential for healthy hair?	219 (68.4)	38 (11.9)	63 (19.7)	36 (36.4)	26 (26.3)	37 (37.4)	183 (82.8)	12 (5.4)	26 (11.8)	**<0.001**
Reading instructions for DS use are unnecessary?	35 (10.9)	237 (74.1)	48 (15.0)	12 (12.1)	71 (71.7)	16 (16.2)	23 (10.4)	166 (75.1)	32 (14.5)	0.81
Too much vitamin intake is harmful to health?	160 (50.0)	66 (20.6)	94 (29.4)	39 (39.4)	26 (26.3)	34 (34.3)	121 (54.8)	40 (18.1)	60 (27.1)	**0.04**
Taking DS may cause kidney disease?	168 (52.5)	31 (9.7)	121 (37.8)	49 (49.5)	9 (9.1)	41 (41.4)	119 (53.8)	22 (10.0)	80 (36.2)	0.67
Taking over-the-counter DS may be dangerous to health?	238 (74.4)	29 (9.1)	53 (16.6)	68 (68.7)	11 (11.1)	20 (20.2)	170 (76.9)	18 (8.1)	33 (14.9)	0.29
DS may be eaten instead of food?	21 (6.6)	258 (80.6)	41 (12.8)	7 (7.1)	71 (71.7)	21 (21.2)	14 (6.3)	187 (84.6)	20 (9.0)	**0.009**
DS have harmful side effects?	134 (41.9)	64 (20.0)	122 (38.1)	42 (42.4)	23 (23.2)	34 (34.3)	92 (41.6)	41 (18.6)	88 (39.8)	0.52
For DS use, periodic blood tests are necessary?	260 (81.3)	17 (5.3)	43 (13.4)	69 (69.7)	10 (10.1)	20 (20.2)	191 (86.4)	7 (3.2)	23 (10.4)	**0.001**
Taking DS with doctor’s instructions promotes body’s health?	280 (87.5)	7 (2.2)	33 (10.3)	84 (84.8)	3 (3.0)	12 (12.1)	196 (88.7)	4 (1.8)	21 (9.5)	0.59

Note: data represented as N (%). *p*-value calculated for DS users in the two sexes. *p* significant at <0.05 level (bold).

**Table 5 ijerph-17-03515-t005:** Source, reason and form of DS use and intake among participants.

Parameters	All DS Users (320)	Male DS Users (99)	Female DS Users (221)	*p*-Value
**Reason for DS Use**				
Hair condition	100 (31.3)	17 (17.2)	83 (37.6)	**<0.001**
Nail health	65 (20.3)	13 (13.1)	52 (23.5)	**0.021**
Body building	56 (17.5)	35 (34.4)	21 (9.5)	**<0.001**
Disease prevention	61 (19.1)	22 (22.2)	39 (17.6)	0.21
Reinforce health	105 (32.8)	32 (32.3)	73 (33.0)	0.51
Vitamin deficiency	187 (58.4)	47 (47.5)	140 (63.3)	**0.006**
**Preferred form of DS intake**				
Capsules	57 (17.8)	16 (16.2)	41 (18.6)	0.364
Pills	144 (45.0)	39 (39.4)	105 (47.5)	0.110
Powder	20 (6.3)	15 (15.2)	5 (2.3)	**<0.001**
Liquid	38 (11.9)	13 (13.1)	25 (11.3)	0.384
**Purchase source**				
Internet	34 (10.6)	22 (22.2)	12 (5.4)	**<0.001**
Pharmacy	125 (39.1)	31 (31.3)	94 (42.5)	**0.034**
Social media	7 (2.2)	4 (4.1)	3 (1.4)	0.136

Note: data represented in N (%). *p*-value calculated for different sexes in DS users. *p* significant at <0.05 level (bold).

**Table 6 ijerph-17-03515-t006:** Logistic regression investigating independent predictors of dietary supplement (DS) use in Saudi adolescents.

Parameters	Crude		Multivariate Adjusted	
	Odd Ratio (95% CI)	*p*-Value	Odd Ratio (95% CI)	*p*-Value
Gender				
Female	Ref:		Ref:	
Male	0.44 (0.340–0.58)	**<0.001**	0.45 (0.320–0.63)	**<0.001**
BMI				
Normal	Ref:		Ref:	
Overweight	0.75 (0.531–0.07)	0.112	0.74 (0.491–0.13)	0.171
Obese	0.90 (0.531–0.52)	0.697	1.06 (0.581–0.92)	0.855
Marital Status				
Married	Ref:		Ref:	
Unmarried	0.73 (0.351–0.52)	0.398	0.79 (0.321–0.92)	0.599
Divorce	1.56 (0.445–0.63)	0.491	1.43 (0.296–0.98)	0.657
Family Income				
<5000	Ref:		Ref:	
5001–10,000	1.86 (0.854–0.02)	0.117	1.73 (0.714–0.24)	0.227
100,01–15,000	2.04 (0.964–0.29)	0.062	1.70 (0.733–0.96)	0.218
>15,000	2.42 (1.214–0.86)	**0.012**	2.44 (1.115–0.33)	**0.025**
Educational Level				
Primary	Ref:		Ref:	
Secondary	1.08 (0.831–0.40)	0.577	1.41 (0.892–0.24)	0.137
Physical Activity				
Yes	1.73 (1.172–0.55)	**0.005**	1.34 (0.832–0.18)	0.236
No	Ref:		Ref:	
If Yes, then types of activity				
Vigorous PA				
No	Ref		Ref:	
Yes	1.36 (1.051–0.76)	**0.021**	0.92 (0.671–0.27)	0.624
High PA				
No	Ref	**0.026**	Ref:	**0.004**
Yes	0.73 (0.550–0.96)		0.62 (0.440–0.86)	
Moderate 1 PA				
No	Ref	0.389	Ref:	0.630
Yes	1.12 (0.861–0.46)		0.93 (0.681–0.27)	
Moderate 2 PA				
No	Ref	0.114	Ref:	0.947
Yes	1.24 (0.951–0.63)		1.01 (0.731–0.39)	
Light PA				
No	Ref	0.699	Ref:	0.375
Yes	0.95 (0.721–0.25)		0.86 (0.631–0.19)	

Note: multivariate model adjusted for potential confounding variables. Values significant at *p* < 0.05 level (bold).

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
