# Peer review of "The Prevalence and Awareness Concerning Dietary Supplement Use among Saudi Adolescents"

_ijerph, 2020, doi:10.3390/ijerph17103515_

Round 1

Reviewer 1 Report

The study has basically a descriptive nature, while the study may include interesting data.

  1. Many readers want to know about whether the DS use is good or bad. This is importantly associated with interpretation of the study results (i.e., the meaning of low and high prevalence).
  2. Why must adolescence and children be examined in the DS use? More description is necessary than the current description (except for limited and inconsistent reports). For instance, can the vitamin intake in the period of adolescence and children lead to a long-span and high QOL life?
  3. Many readers want to know about why the DS is used and why there is a considerable difference in the use among countries. This is importantly associated with the mechanistic explanation of the study results.
  4. Many readers want to know about which variables among many variables were strongly predictive to the DS use. The multivariate models should be statistically adopted. Confounding and correlated conditions could exist in this multi-variables.
  5. How should the results be used in preventative and public strategies?
  6. The study limitations should be described.
  7. The professional editing and native check should be performed (e.g., Current Dietary …; ‘D’ should be changed to ‘d’ [in Abstract], ‘Data was’ should be changed to ‘Data were’ [in Methods]).

Reviewer 2 Report

The introduction is too short; Lack of reference to the cultural elements of the studied phenomenon.
The goal has not been achieved. The study is only a presentation of results, just a simple group comparison. Pity that the authors didn't use statistical modeling and didn't look for relationships between parameters. The authors must describe the exact procedure for selecting a research group, among others - how participants were recruited for research, what the drawing procedure was like.
The weakness of the article is research tool - author's questionnaires.  Questionable measurement of physical activity. Why did the authors not use some standardized tool? They should explain this choice and accurately describe the tool. Figure 1 is  unnecessary

Round 2

Reviewer 1 Report

The revised version has been much improved. A minor revision is still needed.

  1. The authors stated that the DS use is related to healthy lifestyles. For all readers, the authors can add describing whether this DS use is independently related to health outcomes or its combined effects with healthy lifestyles are mainly related to health outcomes. Are there any references or authors’ ideas?
  2. In row (line) 43, ‘DS’ can be fully spelled out and then abbreviated (the abstract and text should be separately independent).
  3. In row 53, ‘US’ can be fully spelled out and then abbreviated (the abbreviation is possible in raw 58).
  4. In row 58-68, the prevalence of ref 30 (Australia) can be included in Introduction.
  5. In row 70, can the sentence cite the ref [7, 8]?
  6. In row 73, ref [22, 7] can be changed to [7, 22].
  7. In row 82, the actual prevalence (%) or data by age can be concretely described in Saudi Arabia.
  8. In Table 6, the row slipped in terms of the ‘type of activity’.
  9. The words ‘use’ and ‘usage’ were mixed in the text.

Reviewer 2 Report

The paper improved by the authors is reliable prepared. Thank you.
